# The genetic trail of the invasive mosquito species *Aedes koreicus* from the east to the west of Northern Italy

Laura Soresinetti[1], Giovanni Naro[1], Irene Arnoldi[1], Andrea Mosca[2], Katja Adam[3], Heung Chul Kim[4¤a], Terry A. Klein[4¤b], Francesco Gradoni[5], Fabrizio Montarsi[5], Claudio Bandi[1,6], Sara Epis[1,6]*, Paolo Gabrieli[1,6]*

1 Department of Biosciences and Pediatric Clinical Research Center "Romeo Ed Enrica Invernizzi", University of Milan, Milan, Italy, 2 Istituto per le Piante da Legno e l'Ambiente, I.P.L.A. S.p.A., Turin, Italy, 3 Department of Biodiversity, Faculty of Mathematics, Natural Sciences and Information Technologies, University of Primorska, Koper, Slovenia, 4 Force Health Protection and Preventive Medicine, Medical Department Activity-Korea/65th Medical Brigade, United States Army Garrison-Humphreys, Pyeongtaek, South Korea, 5 Istituto Zooprofilattico Sperimentale delle Venezie, Legnaro (Padua), Italy, 6 Italian Malaria Network, Inter University Center for Malaria Research, University of Camerino, Camerino (Macerata), Italy.

¤a Current address: U Inc., Seoul, Republic of Korea
¤b Current address: PSC 450, Box 75, APO AP 96206, Pyeongtaek, United States of America
* sara.epis@unimi.it (SE); paolo.gabrieli@unimi.it (PG)

## Abstract

### Background

*Aedes koreicus* is native to Far East Asia and recorded in Europe since 2008. In Italy, *Ae. koreicus* is widespread throughout the Northern part of the peninsula, highlighting its invasive potential and spread. However, no clear clues about the dispersal patterns of the species have been collected so far.

### Methodology/Principal findings

Population genetic analyses were performed to assess the genetic structure of populations of *Ae. koreicus* and to make hypotheses about its dispersal patterns in Northern Italy. Ten microsatellite markers specific for *Ae. koreicus* were used to genotype 414 individuals from 13 populations in the pre-alpine area of Italy, and neighboring Slovenia. Basic and Bayesian population genetic analyses were performed to evaluate patterns of genetic variation, genetic structure, and demography of selected mosquito populations. While presenting a certain degree of structuring, the Italian and Slovenian populations of *Ae. koreicus* were poorly differentiated. Moreover, demographic analysis supports the expansion of a single population propagule of *Ae. koreicus* in Italy and Slovenia and provides evidence of the presence of overwintering populations in the studied area.

### Conclusions/Significance

Our results highlight a common origin, and stable colonization of Northern Italy and Slovenia, as a probable consequence of the expansion of a unique population. This stresses

**Data availability statement:** All data and results of the analyses are available within the manuscript or in the Supporting Information files, except for raw genotypes of individual mosquitoes included in the analyses of the present study that can be accessed on Zenodo at the following link: doi.org/10.5281/zenodo.12800346.

**Funding:** The present study was done under the framework of the Project PE-13, INFACT, RESEARCH NODE2 Arthropod Vectors and Vector-Borne Diseases, which is part of the National Recovery and Resilience Plan (NRRP; Project number PE00000007, to CB, SE, PG), and the Armed Forces Health Surveillance Division, Global Emerging Infections Surveillance (GEIS) Branch, ProMIS ID P0016_21_ME to TAK and HCK. None of the funding sources had roles in study design, collection, analysis, and interpretation of data, in the writing of the report, and in the decision to submit this article for publication.

**Competing interests:** The authors have declared that no competing interests exist.

out the importance of continuous monitoring of *Ae. koreicus*, to finally uncover the geographic origins and entrance pathways of invasive populations and to prevent or limit further introductions.

## Author summary

In the present manuscript, the population genetics of the invasive mosquito *Aedes koreicus* is presented. Specifically, 13 invasive (Italy, Slovenia) and one native (Republic of Korea, ROK) populations have been analyzed, with a particular focus on individuals from Northern Italy, an area where a widespread presence of the mosquito species has been observed. Results revealed a low variability in populations from non-endemic countries, even when collected in different years. A stable colonization is thus highlighted. This is probably the consequence of a single event of importation, as supported by demographic analysis. Individuals from the ROK mainland here analyzed are not the source of this introduction, being genetically unrelated to the Italian and Slovenian individuals. This research provided novel data for the investigation of the pattern of expansion of *Ae. koreicus* in Northern Italy, and thus pose important basis for future studies of the invasive corridors of the species to limit/prevent its further dispersion.

## Introduction

*Aedes koreicus* (Edwards 1917) (Diptera: Culicidae), the Korean bush mosquito, is native to Eastern Russia (Primorye region), Northern China, and the Korean Peninsula [1]. The invasive potential of *Ae. koreicus* was first documented after its first detection in Europe, namely in Belgium in 2008 [2], where it is now established. From that time, the species has been recorded in 10 additional countries of Central and Eastern Europe [3]. Among those, the Italian invasion is of particular interest. *Aedes koreicus* was detected in Italy in 2011 [4], and within a decade it has been recorded in the whole pre-alpine area of the peninsula [5–7]. The widespread dispersal of *Ae. koreicus* in Italy is an exception to the invasive pattern observed in other European countries, where the mosquito populations are confined to small areas close to the site of first introduction [8]. Notably, in the area where *Ae. koreicus* is endemic, two morphotype have been described: a first one that is generally detected in the mainland of the Republic of Korea (ROK), and a second one that is prevalent in Jeju Island (between the ROK mainland and Japan) [9–11]. These two morphotypes are distinguishable by the absence of a band on the hind tarsomere V in the individuals collected in the ROK mainland, present instead in the mosquitoes diffused in the Jeju Island. In all the countries invaded by *Ae. koreicus* (including Italy) only the morphotype of Jeju Island have been found; exception to this is the population from the municipality of Wiesbaden, in Germany [2,5,6,12].

*Aedes koreicus* represents a potential threat to human and animal health in Europe, because of its possible vectorial competence for viruses and nematodes [13–18]. Indeed, individuals of *Ae. koreicus* positive for *Dirofilaria repens* have recently been collected in Hungary [17]. In addition, experimental studies have provided some evidence for the capability of these mosquitoes to transmit chikungunya virus, Sindbis virus, and, at lower rate, Zika virus [13,15,16]. However, no evidence has so far been reported for a contribution of *Ae. koreicus* to arboviral outbreaks in Europe [19].

The development of control measures to limit *Ae. koreicus* proliferation is anyway needed, since only a limited number of studies have thus been devoted to this issue. In this context, a

recent publication demonstrated the efficacy on *Ae. koreicus* larvae of a bioinsecticide based on *Bacillus thuringiensis* [20].

Besides the use of direct control measures, such as larvicides, another main pillar in the control of invasive mosquitoes is the identification of the pathways of their dispersal. This knowledge is required to design appropriate surveillance strategies and activate proper control measures in non-infested areas [21].

In the case of *Ae. koreicus*, it remains unclear how it spread in Italy. Human-mediated transportation is likely to play a major role in dispersal patterns, a phenomenon that was described shaping the dispersal of other invasive species of mosquitoes [22,23]. A previous study revealed a large-scale dispersal of *Ae. koreicus* in Europe [24], but the events at the basis of this distribution pattern still need to be elucidated. Population genetics, e.g., population structure and phylogeography, can provide useful insights into the invasion corridors and the dispersal patterns of invasive mosquito species [25,26]. This information can help in preventing the further expansion of *Ae. koreicus,* which was predicted both in Europe and worldwide [5,11,27].

In the present study, microsatellite markers were used to perform population genetics of *Ae. koreicus*, with a particular focus on the populations dispersed throughout the pre-alpine area of Northern Italy. Through comprehensive population genetic analyses, we aimed at elucidating the genetic relationship and the invasion dynamics in Italy.

## Materials and methods

### Populations and samples details

A total of 414 *Ae. koreicus* specimens from 10 disparate collection sites were investigated (Table 1 and S1 Table). Specifically, we define as "population" groups of individuals of *Ae. koreicus* collected in different breeding sites from the same municipality in the same year. In Fig 1, a map representative of the distribution of these populations is reported. Notably, the

**Table 1. Details of *Ae. koreicus* collected from Italy, Slovenia, and the Republic of Korea (ROK) that were used in the population genetic study. For each population, details about the first historical record, year of collection, symbol for the identification, and number of samples are indicated (N). For the historical reports, the references are indicated. Geographical references of the collection site and details of single specimens can be found in S1 Table.**

| Country (First historical record) | Area | First historical report in the district | Population | Year of collection | Symbol | N[a] |
|---|---|---|---|---|---|---|
| Italy (2011) [4] | North-East of Italy | 2011 [4] | Belluno | 2011 | BL11 | 24 |
| | | | | 2021 | BL21 | 31 |
| | | 2012 [6] | Vicenza | 2021 | VI21 | 30 |
| | | | | 2022 | VI22 | 30 |
| | North-Central Italy | 2013 [29] | Como | 2021 | CO21 | 30 |
| | | 2015 [30] | Sondrio | 2021 | SO21 | 30 |
| | | | | 2022 | SO22 | 30 |
| | | 2020 [31] | Trescore Balneario (Bergamo) | 2022 | TR22 | 30 |
| | | 2021 [5] | Fonteno (Bergamo) | 2021 | FO21 | 30 |
| | | | | 2022 | FO22 | 30 |
| | | 2021 [5] | Brescia | 2021 | BS21 | 30 |
| | North-Western Italy | 2012 [7] | Asti | 2021 | AT21 | 30 |
| Slovenia (2013) [32] | North-Western Slovenia | 2021 (present study) | Ajševica (Nova Gorica) | 2021 | SL21 | 30 |
| ROK (Native, 1920) [9] | Wester ROK | / | Pyeongtaek (Gyeonggi) [b] | 2021 | KO21 | 29 |

[a]N, total number of individual mosquitoes analysed for each population.

[b]Specimens were collected at the US Army Garrison, Humphreys, located adjacent to Anjeong-ri, Pyeongtaek city, as part of a malaria and Japanese encephalitis surveillance program.

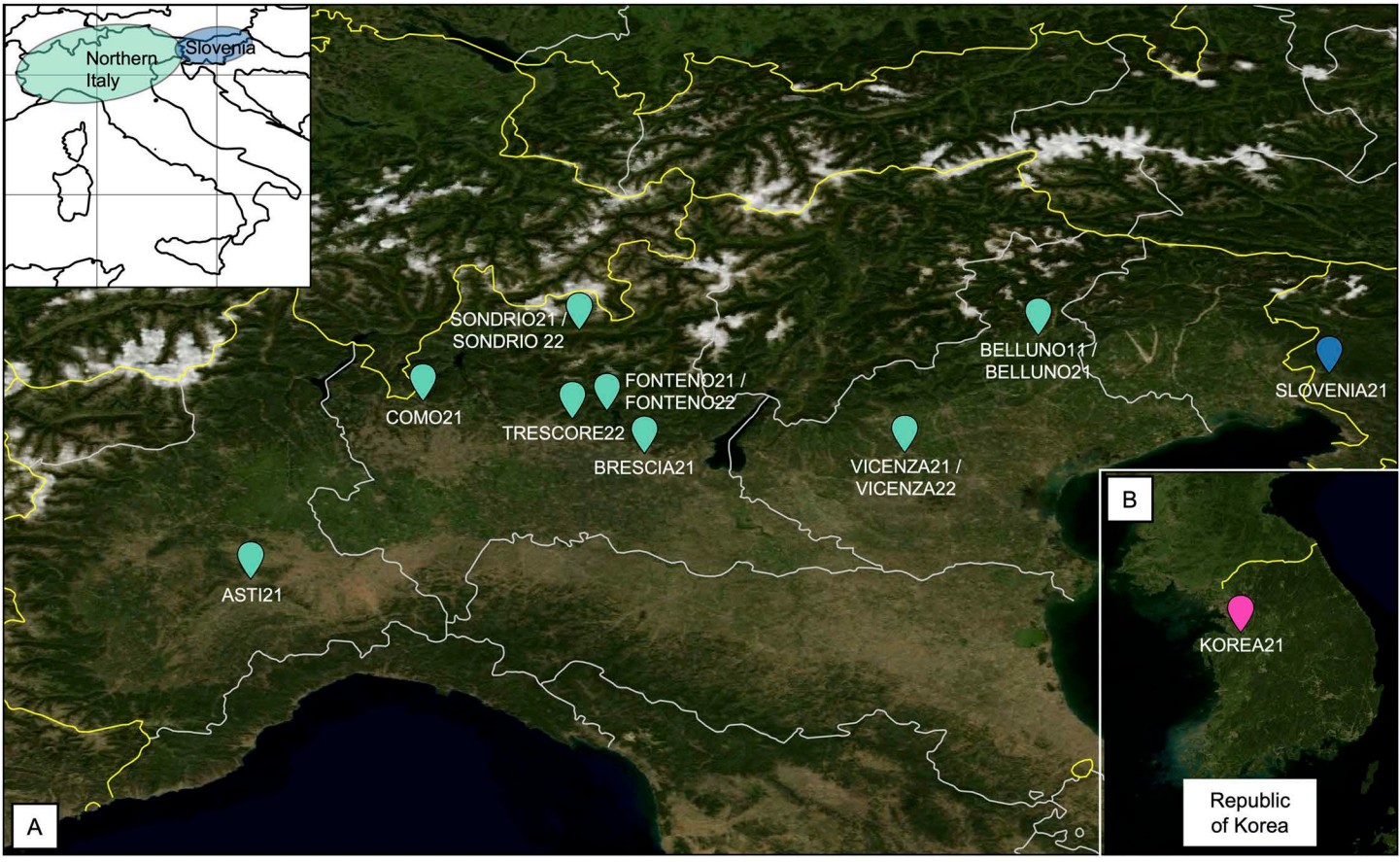

**Fig 1. Populations of *Ae. koreicus* analyzed in the present study.** (A) The map of Northern of Italy and Slovenia is provided, and locations from which individuals of *Ae. koreicus* were collected are indicated in green and blue, respectively. (B) In pink, population from the native area, i.e., ROK, is indicated. The map was generated with QGIS [28]. The satellite images were available at the "MODIS Blue Marble, Next Generation" Dataset from NASA Earth Observatory (https://visibleearth.nasa.gov/images/74117/august-blue-marble-next-generation), and obtained thanks to the NASA Worldview application (https://worldview.earthdata.nasa.gov) which is part of the NASA Earth Science Data and Information System (ESDIS) (used based on the NASA policies https://visibleearth.nasa.gov/image-use-policy). Country and regional border shapes have been downloaded from the "Natural Earth" public Database (https://www.naturalearthdata.com/downloads/10m-cultural-vectors/ maps: Admin_0_land_boundaries and Admin_1_States_provinces_lines. https://www.naturalearthdata.com/downloads/50m-cultural-vectors/ map: Admin_0_boundary_lines_land; https://www.naturalearthdata.com/downloads/50m-physical-vectors/ map: coastline).

analyses were done on a total of 14 populations since for four locations, samples collected in the same sites in different years were included in the study.

Twelve populations were collected in eight districts located in four different regions of Northern Italy: Belluno and Vicenza districts in the Veneto region; Como, Sondrio, Bergamo (two localities, Fonteno and Trescore Balneario), and Brescia districts in the Lombardy region; Asti district in the Piedmont region. Individuals of *Ae. koreicus* representative of these areas were collected as previously described [5,6]. In brief, larval breeding sites were monitored, and larvae and pupae of mosquito were collected and brought to the insectarium to allow the emergence of adult mosquitoes. A maximum of three accesses for breeding site, spanning in a time frame of maximum four weeks, were performed to assure the collection of a sufficient number of samples for each population (see S1 Table for details about the dates of collection of single individuals).

*Aedes koreicus* mosquitoes from Slovenia were collected as adults in Ajševica, a settlement located in the municipality of Nova Gorica, and close to the Eastern Italian border. Adult

mosquitoes from the native area of *Ae. koreicus* were collected at US Army Garrison Humphreys located adjacent to Anjeong-ri, Pyeongtaek city, Gyeonggi province, ROK.

*Aedes koreicus* field collections were conducted during 2021, except for Trescore Balneario where individuals were collected in 2022 (Table 1). Additionally, two populations were examined from Belluno district; one of these was sampled in 2011, i.e., the year of the first finding of *Ae. koreicus* in Italy [4], and the second in 2021. To assess the presence of overwintering populations of *Ae. koreicus*, specimens from Fonteno, Sondrio, and Vicenza were analyzed and compared for two consecutive years (2021-2022).

From the morphological point of view, all specimens that were collected in Northern Italy and Slovenia showed a morphotype typical of those diffused in Jeju Island (ROK), while specimens from the ROK mainland displayed the continental morphotype. This is in concordance to what described in literature [9–11].

Adults of *Ae. koreicus* resulting from the samplings done in 2021 in the Lombardy and Piedmont regions (Italy) and in Slovenia were stored in 95% EtOH and maintained at -20°C until DNA extraction. All the samples collected in the Veneto region, together with the mosquitoes collected in the Lombardy region in 2022 and specimens collected from the ROK, were stored dry at -80°C before proceeding with the extraction of DNA.

## DNA extraction, and PCR amplification of microsatellite markers

DNA was extracted from individual *Ae. koreicus* using the DNeasy Blood & Tissue kit (Qiagen, Hilden, Germany) and used as template DNA for polymerase chain reaction (PCR) to recover microsatellite data. Specifically, in S2 Table the panel of 11 specific short sequence repeats (SSRs), and the primer pairs for their detection are reported; SSRs were amplified in a multiplex assay, accordingly to the protocol described in [33]. Amplicons were loaded on 1.5% agarose gels to verify the amplification, and all correctly amplified samples were sent to Eurofins Genomics (Ebersberg, Germany) for capillary electrophoresis as fragment length analysis (FLA service). Each electropherogram was used to deduce the lengths of the alleles of SSRs *loci* for each specimen. The resulting raw dataset was deposited in Zenodo at the following link: doi.org/10.5281/zenodo.12800346.

## Relatedness analysis, SSRs features, within and among population genetic variability

Relatedness of *Ae. koreicus* individuals composing each population was evaluated through the estimation of kinship coefficients (*k*). The kinship coefficient corresponds to "ratios of differences of probability of identity in state" from homologous *loci* randomly sampled from each pair of individuals and thus provides an index of the relative relatedness of all genotyped individuals in a data set [34]. SPAGeDi program [35] was used for the computations of *k* values as described by Loiselle and collaborators (1995) [36]; specifically, *k* estimations were performed among pairs of individuals collected from the same breeding sites or sampled in breeding sites located at ≤ 1km (see S1 Table for collection coordinates of single specimens). Samples were considered as full siblings when *k* was >0.1875, while putative half-siblings were identified when 0.0938> *k* >0.1875. These intervals were proposed by Iacchei and colleagues (2013) [34] and were tested in other studies analyzing mosquito species [37,38]. A second estimation of relatedness level was performed using the software ML-Relate [39]. Half or full siblings according to both computations (i.e., output of SPAGeDi and output from ML-Relate) were considered as actual siblings, and just one of them was included in further population genetics analyses.

Patterns of Linkage Disequilibrium (LD) for each couple of *loci* across all populations were estimated using the online software GenePop v. 4.3 [40,41], and Bonferroni correction was applied to the resulting p-values.

Polymorphic Information Content (PIC) values associated with the SSRs *loci* were then estimated using the software Cervus v. 3.0.7 [42]. This software was used also to verify eventual deviation from the Hardy–Weinberg equilibrium (HWE). The HWE was evaluated at each *locus*/sample combination for specimens from each population using Fisher's exact test, followed by Bonferroni correction.

Genetic variability of specimens composing each population was assessed estimating the allelic richness (Ar) and the number of private alleles (Np) in R using the packages *HierFstat* v0.5-11 and *poppr* v2.9.4 in R, respectively [43,44]. Observed and expected heterozygosities (Ho and He) were also determined using *HierFstat* v0.5-11 [43].

The inbreeding coefficient within individuals of the same population ($F_{IS}$), was calculated with *HierFstat* v0.5-11 in R [44]. For each value, 10000 bootstraps over all *loci* were computed to assess 95% confidence intervals.

Another fixation index ($F_{ST}$), was estimated to assess the variability among different populations of *Ae. koreicus*. Pairwise $F_{ST}$ ($F_{ST}P$) values were computed to evaluate the genetic distance between pairs of populations using *HierFstat* v0.5-11 in R [43] with the function "genet. dist" and setting "Nei87" as computational method [45].

$F_{ST}Ps$ on paired population VI21-VI22, SO21-SO22, FO21-FO22 were used to investigate the presence of *Ae. koreicus* overwintering populations in Northern Italy. A Wilcoxon Signed-Rank test was used to assess the statistical significance of the comparison between $F_{ST}P$ values computed for potentially overwintering populations and $F_{ST}Ps$ of populations collected in the same year from different locations.

The Mantel test [45] was performed using GenAlEx 6.5 [46] to evaluate the subsistence of isolation by distance phenomena. Particularly, 1000 permutations analyses were performed to compare the population $F_{ST}P$ matrix with the pairwise geographical distance matrix computed using UTM coordinates (GGD). Specifically, median coordinates representative of all the collection sites from which single individual of a population have been collected are reported in S1 Table. In addition, for populations sampled from the same location during more than one season (i.e., BL11 and BL21; VI21 and V22; SO21 and SO22; FO21 and FO22), only the population composed by individuals collected during 2021 was retained to minimize eventual biases in consequence of the sampling season. KO21 was excluded because of strong genetic differences among individuals within this population and other invasive populations, and because of its great geographical distance.

## Population structure and co-ancestry analyses

To evaluate pattern of genetic structuring, the Principal Component Analysis (PCA) and Discriminant Analysis of Principal Components (DAPC) were computed using the R package *ade4* v 1.7-22 [47] and plotted with ggplot2 v3.4.4 [48]. These analyses were performed on two subsets of the populations: one considering the 14 populations of *Ae. koreicus* previously described, and one including only the six populations analyzed to assess overwintering (i.e., VI21-VI22; SO21-SO22; FO21-FO22).

To additionally evaluate the genetic structure and to define the degree of ancestry characterizing each population of *Ae. koreicus* Bayesian analysis was performed using the software STRUCTURE v 2.3.4 [49]. The STRUCTURE analysis was computed for 11 populations: in case of populations that were sampled for two consecutive years (i.e., Vicenza, Sondrio, Fonteno), only individuals collected during 2021 were considered. Inferences about the populations of *Ae. koreicus* were done under the admixture model and assuming correlated frequencies of the alleles. 500,000 burn-in and 1,000,000 Monte Carlo Markhov Chain (MCMC) replicates were tested, while the K interval was set between 1 and 11 (i.e., the total number of populations analyzed); for each K, 20 independent runs were iterated. The results were further analyzed using the package

STRUCTURE HARVESTER [50], and the most probable number of K was determined considering the log probability L(K) and ΔK over 20 runs [51]. Once the most descriptive K was identified, the Greedy algorithm implemented in the program CLUMPP [52] was used to align and merge the 20 runs resulting from STRUCTURE computations. In this way, the percentage of cluster membership coefficients (K) could be estimated for each specimen. Lastly, DISTRUCT [53] was used to plot the relative co-ancestry representation.

## Demographic analysis of the populations of *Ae. koreicus*

The standard Approximate Bayesian Computation (ABC) methodology (ABC-LDA) implemented in the software DIYABC v2.1 [54] was used to compare various demographic scenarios and thus identify the most descriptive one of the patterns of population divergence during the invasion of *Ae. koreicus* of Northern Italy and Slovenia.

Ten populations of *Ae. koreicus* were considered in the DIYABC analysis. As done for STRUCTURE, only individuals collected in 2021 were considered for Vicenza, Sondrio, and Fonteno populations. KO21 was excluded because it introduced strong biases which resulted in the decrease of the power of resolution of the Bayesian computations. Indeed, the main purpose of the present demographic analysis was to verify if populations dispersed in Italy and Slovenia were derived by the introduction and expansion of a unique propagule. Nine possible invasive scenarios were designed on the basis of first historical records for the introduction of *Ae. koreicus* in invaded areas together with the data of previous population genetic analyses (i.e., genetic variability, population structure, and ancestry), and then tested to assess the confidence of the expansion hypothesis (S1 Fig).

Once competing scenarios were designed, information about historical and genetic models are needed by DIYABC to perform computations. The genetic variation of populations was defined using the following statistics: the mean number of alleles per *locus*, the mean expected heterozygosity, the mean allelic size variance, $F_{ST}P$ values, and the Garza-Williamson index (i.e., number of alleles in a population divided by the range in allele size, which helps in discriminating recently and stably introduced populations) [55].

Concerning the historical models, priors and parameters of demographic distribution are reported in Table 2. Because of the relatively recent reports of the introduction of *Ae. koreicus* and of the poor *a priori* knowledge, all parameters were kept as broad as possible. The effective population size, the timing in which founding events occurred, the number of founder individuals that contributed to the establishment and expansion of invasive populations, the duration of bottleneck events and the rate of admixture were taken in consideration as priors; for all, a uniform distribution was considered. N range was adjusted according to in-field observations for the dimension of the natural populations; for this reason, four different ranges were established: 10-10000 for NA (= unknown ancestral population); 10-8000 for BL11, BL21, and VI21; 10-1000 for SO21; and 10-5000 for all the other populations. Time range (t) of each founding event, was set considering the historical data of the recent importation of *Ae. koreicus* in Italy and Slovenia. However, since a level of uncertainty could affect the dates of first detection, three overlapping intervals were used: 5-50 was set for founding events occurring in the last 5 years; 10-70 was used as interval for events occurring in a period of ~10 years; 10-100 was instead set to define the direct derivation from an ancient unidentified population. A similar level of uncertainty was considered for the duration of eventual bottleneck events, for which a period comprised between 1 and 50 was set. The number of founder individuals for each colonization event was described as NF. Similarly for N and t, three intervals were fixed taking into consideration in-field observations: 10-1000 for BL11, BL21, VI21; 10-300 for SO21; 10-500 for all the other populations.

**Table 2. Prior distribution and values of parameters used for DIYABC analyses.**

| Parameter | Distribution | Values |
|---|---|---|
| Effective population size (N) | Uniform | |
| NA[a] | | 10-10000 |
| $N_{BL11}$, $N_{BL21}$, $N_{VI21}$ | | 10-8000 |
| $N_{CO21}$, $N_{TR22}$, $N_{FO21}$, $N_{BS21}$, $N_{AT21}$, $N_{SL21}$ | | 10-5000 |
| $N_{SO21}$ | | 10-1000 |
| Number of founder events (NF) | Uniform | |
| $N_{BL11}$, $N_{BL21}$, $N_{VI21}$ | | 10-1000 |
| $N_{CO21}$, $N_{TR22}$, $N_{FO21}$, $N_{BS21}$, $N_{AT21}$, $N_{SL21}$ | | 10-500 |
| $N_{SO21}$ | | 10-300 |
| Bottleneck duration (db) | Uniform | 1-50 |
| Time of founder events | Uniform | |
| ta, tb: founder events occurred within ~5 years | | 5-50 |
| tc, td: founder events occurred within ~10 years | | 10-70 |
| te: direct derivation from NA | | 10-100 |
| Rates of admixture | Uniform | 0.001-0.999 |
| *Microsatellite mutation model* | | |
| Mean mutation rate | Gamma | $10^{-6}$-$10^{-3}$ |
| Mean parameter of the geometric distribution of repeats (P) | Gamma | 0.1-0.3 |
| Mean single nucleotide insertion/deletion mutation rate (SNI) | Log-uniform | $10^{-8}$-$10^{-5}$ |

[a]NA, unknown ancestral population

To evaluate the concordance with observed data, one million simulated datasets were generated for each scenario. These were compared through the calculation of their posterior probabilities as a polychotomous logistic regression on 1% of simulated data closest to the observed data, while confidence in the choice of the most probable scenario was evaluated with the computation of type I and type II errors [56,57]. Based on these parameters and estimations, the most probable scenario was identified, and the posterior distributions of genetic and demographic parameters were estimated with a local linear regression on 1% of the simulated data closest to our observed dataset, after applying a logit transformation to the parameters value. As last step of validation, one thousand pseudo-observed data sets were computed using parameter values drawn from the posterior distribution given the most likely scenario to perform a model checking evaluation.

## Results

A total of 414 *Ae. koreicus* from 13 populations in Italy and Slovenia and one population from its endemic area, i.e., in the ROK [1], were evaluated for genetic variability characterizing the populations using the SSRs specifically developed for the Korean bush mosquito (S2 Table) [33].

Once all the 4554 electropherograms (11 *loci* amplified in 3 multiplexes for 414 individuals) were analyzed, the allelic lengths were rounded to the integer prior to population genetic analysis. The analysis of 10 SSRs was successful, and the complete genotyping of most of *Ae. koreicus* individuals was achieved (genotypes of individuals rounded to the integers are reported in S1 File). Unique exception resulted the *locus* AK03 that was characterized by a high difference between He and Ho and by numerous incomplete genotypes, suggesting the presence of null alleles. Because of this, the AK03 *locus* was excluded to avoid biases in further population genetics analysis.

Estimation of kinship coefficients demonstrated that all the populations of *Ae. koreicus* analyzed were characterized by the presence of related individuals (S3 Table), with percentages ranging from 5.82% (CO21) to 20.30% (BS21). The highest number of full siblings (FS) was detected in TR22 and FO22 (31 FS pairs for both populations). Conversely, the highest number of half siblings (HS) was observed in SO22 (55 HS pairs). One FS or HS individual was retained in the dataset used for population genetics analysis: 201/414 *Ae. koreicus* individuals were included in the analyses described in the following paragraphs (S1 File, individuals in bold).

## SSR markers characteristics and intra-population variability

No signals of LD were observed among the 10 SSRs used to assess population genetics. Concerning HWE, departures from the equilibrium were observed in one *locus* of BS21, i.e., AK28 (p-values = 0.002). This departure is a consequence of an excess of observed heterozygosity, since Ho = 1 (S4 Table).

Basic statistics computed to assess intra-population variability are displayed in Table 3. Indices were reported as average values characterizing each population of *Ae. koreicus*. Details about statistics of each *locus* in single populations are shown in S4 Table. PIC mean value was equal to 0.344. Alleles of the AK13 and AK15 *loci* were fixed in the majority of the populations and, therefore, their PIC values strongly reduced the overall informative content of the SSR markers (S4 Table). Besides the above-mentioned values, He and Ho were comparable among most populations, with overall mean values of 0.396 and 0.399, respectively (Table 3).

Mean Ar was 2.796 and the number of alleles was strongly influenced by the fixation of AK15 (Table 3). The AK15 *locus* was polymorphic only in BL21 and TR22. Populations with the highest values of allelic richness were BL11, BL21, SO22, TR22, and KO21 (for all, Ar > 2.9), which were also characterized by the presence of private alleles. Specifically, the highest number of Np were found in several individuals from the sample specimens from the ROK (Np = 55, in AK05, AK07, AK09, AK13, AK28) and from TR22 (Np = 7, distributed in AK10, AK12, AK15) (S4 Table).

**Table 3. Intra-population variability among 14 *Ae. koreicus* populations. Number of individuals (N), mean values of allelic richness (Ar), number of private alleles (Np), mean observed (Ho) and expected (He) heterozygosities, mean Inbreeding coefficients ($F_{IS}$) and 95% confidence intervals (95%CI), and mean Polymorphic Information Content (PIC). In addition, mean values of each index are reported (lane "Total").**

| Pop | N | Ar | Np | PIC | Ho | He | $F_{IS}$ (95% CI) |
|---|---|---|---|---|---|---|---|
| BL11 | 13 | 3.140 | | 0.391 | 0.477 | 0.452 | -0.033 (-0.181-0.083) |
| BL21 | 17 | 3.042 | 2 | 0.343 | 0.418 | 0.385 | -0.056 (-0.189-0.022) |
| VI21 | 17 | 2.784 | | 0.339 | 0.376 | 0.395 | 0.071 (-0.104-0.190) |
| VI22 | 15 | 3.091 | 1 | 0.385 | 0.380 | 0.386 | 0.090 (-0.103-0.175) |
| CO21 | 19 | 2.471 | | 0.302 | 0.318 | 0.352 | 0.115 (-0.073-0.295) |
| SO21 | 13 | 2.547 | | 0.348 | 0.385 | 0.416 | 0.051 (-0.044-0.188) |
| SO22 | 14 | 3.089 | 2 | 0.405 | 0.386 | 0.479 | 0.183 (0.066-0.336) |
| TR22 | 12 | 3.255 | 7 | 0.391 | 0.392 | 0.455 | 0.154 (-0.035-0.304) |
| FO21 | 16 | 2.171 | | 0.280 | 0.362 | 0.343 | -0.055 (-0.187-0.062) |
| FO22 | 10 | 2.800 | | 0.317 | 0.440 | 0.391 | -0.118 (-0.376-0.060) |
| BS21 | 13 | 2.777 | | 0.359 | 0.438 | 0.421 | 0.024 (-0.290-0.254) |
| AT21 | 11 | 2.336 | | 0.259 | 0.336 | 0.318 | -0.054 (-0.210-0.107) |
| SL21 | 13 | 2.701 | | 0.355 | 0.446 | 0.417 | -0.065 (-0.209;0.081) |
| KO21 | 18 | 2.934 | 55 | 0.342 | 0.389 | 0.384 | -0.029 (-0.064;0.038) |
| Total | 201 | 2.796 | / | 0.344 | 0.396 | 0.399 | 0.092 |

$F_{IS}$ average values were close to zero for most populations (overall mean = 0.092), indicating the tendency to panmictic condition in *Ae. koreicus* populations analyzed. SO22 ($F_{IS}$ = 0.183; 95%CI: 0.066-0.336) and TR22 (0.154; 95%CI: -0.035-0.304) are notable exceptions and were also both characterized by ~7% of FS (S3 Table). In addition, considering the population pairs collected in two consecutive years, a slight increase of $F_{IS}$ values was observed in populations collected in 2022 for populations from Vicenza and Sondrio districts.

## Genetic variability among different populations

The genetic distance and relationships characterizing the populations of *Ae. koreicus* were assessed computing $F_{ST}P$ values. The resulting matrix can be visualised in Table 4, while 95% confidence intervals derived from 10000 bootstrap are reported in S5 Table. $F_{ST}Ps$ of pairs of populations tested for overwintering (i.e., VI21-VI22, SO21-SO22, FO21-FO22) are highlighted in bold.

Overall, $F_{ST}P$ values varied from 0.000-0.299. According to those, KO21 resulted genetically distant from all the other populations from the invasive area, with the highest diversity among the couples KO21-CO21 and KO21-FO21 ($F_{ST}P$ values equal to 0.299 and 0.296, respectively). This suggests that KO21 likely does not represent the origins of Italian and Slovenian populations of *Ae. koreicus* analyzed in this study.

*Aedes koreicus* populations from Italy and Slovenia were genetically homogeneous. Excluding the pairs of populations considered for overwintering and collected during two consecutive years, the lowest $F_{ST}P$ values were observed among the pairs BL11-VI22 and BL11-SL21 (0.000 for both), BL11-VI21 and TR22-SO21 (0.003 for both computations), and CO21-SO21 (0.004). Even the comparison of the two populations from Belluno (i.e., BL11 and BL21) resulted in a low genetic distance (0.008). In general, BL11 and BL21 populations displayed a high level of genetic similarities with other geographically close populations, i.e., VI21, VI22 and SL21. $F_{ST}P$ values suggested the existence of a genetic relationships among the VI21 population and *Ae. koreicus* collected from the central part of Northern Italy (TR22, BS21, FO22, FO21, SO21, SO22, CO21). Low $F_{ST}P$ values were also detected in pairs of CO21-SO21, SO21-TR22, and FO21-TR22. AT21, which is the unique population from North-Western

**Table 4. $F_{ST}P$ values of the 14 populations of *Ae. koreicus*.**

|       | BL11   | BL21  | VI21  | VI22  | CO21  | SO21   | SO22  | TR22  | FO21  | FO22  | BS21  | AT21  | SL21  | KO21 |
|-------|--------|-------|-------|-------|-------|--------|-------|-------|-------|-------|-------|-------|-------|------|
| BL11  | NA     |       |       |       |       |        |       |       |       |       |       |       |       |      |
| BL21  | 0.008  | NA    |       |       |       |        |       |       |       |       |       |       |       |      |
| VI21  | 0.003  | 0.035 | NA    |       |       |        |       |       |       |       |       |       |       |      |
| VI22  | -0.003 | 0.019 | **0.007** | NA |    |        |       |       |       |       |       |       |       |      |
| CO21  | 0.090  | 0.032 | 0.080 | 0.081 | NA    |        |       |       |       |       |       |       |       |      |
| SO21  | 0.075  | 0.060 | 0.052 | 0.070 | 0.004 | NA     |       |       |       |       |       |       |       |      |
| SO22  | 0.072  | 0.072 | 0.064 | 0.092 | 0.032 | **-0.005** | NA |    |       |       |       |       |       |      |
| TR22  | 0.033  | 0.034 | 0.012 | 0.017 | 0.029 | 0.003  | 0.030 | NA    | 56    |       |       |       |       |      |
| FO21  | 0.106  | 0.138 | 0.052 | 0.086 | 0.113 | 0.052  | 0.073 | 0.028 | NA    |       |       |       |       |      |
| FO22  | 0.072  | 0.091 | 0.028 | 0.067 | 0.067 | 0.007  | 0.048 | 0.010 | **0.006** | NA |  |       |       |      |
| BS21  | 0.034  | 0.041 | 0.020 | 0.029 | 0.049 | 0.050  | 0.057 | 0.027 | 0.072 | 0.049 | NA    |       |       |      |
| AT21  | 0.040  | 0.075 | 0.038 | 0.041 | 0.109 | 0.066  | 0.075 | 0.053 | 0.102 | 0.062 | 0.095 | NA    |       |      |
| SL21  | 0.000  | 0.017 | 0.015 | 0.007 | 0.082 | 0.060  | 0.059 | 0.031 | 0.108 | 0.064 | 0.046 | 0.038 | NA    |      |
| KO21  | 0.201  | 0.250 | 0.228 | 0.228 | 0.299 | 0.274  | 0.264 | 0.222 | 0.296 | 0.248 | 0.231 | 0.265 | 0.240 | NA   |

Italy included in the analyses, resulted genetically more correlated to populations from Belluno and Vicenza.

$F_{ST}P$ was estimated also among pairs of FO21-FO22, SO21-SO22, and VI21-VI22 to evaluate overwintering of established populations of *Ae. koreicus* (Table 4). The observed values among populations collected from the same localities were among the lowest (VI21-VI22: 0.007; SO21-SO22: 0.000; FO21-FO22: 0.006). These $F_{ST}Ps$ were compared with the values obtained by performing the analysis of individuals collected during the same year at different locations (i.e., VI21-SO21, VI21-FO21, SO21-FO21, VI22-SO22, VI22-FO22, SO22-FO22) using a non-parametric test (p-value = 0.026), indicating that the populations from the same location among years were more closely related than the populations collected during the same year from different locations.

The Mantel test was performed to evaluate the influence of geographic distance on the genetic structures and genetic differences among the populations of *Ae. koreicus.* The $R^2$ value resulting from the linear correlation of genetic and geographic distance matrices is 0.0038 (S2 Fig), and therefore it is possible to conclude that no correlation exists among these two variables.

## Analyses of the genetic structure of the populations of *Ae. koreicus*

The high genetic homogeneity of *Ae. koreicus* individuals collected at different locations where it was introduced was confirmed by the plotting of PCA (Fig 2A). The PCA also confirmed the genetic distance of KO21 cohort of *Ae. koreicus* (all in the fourth quadrant, except for few individuals) and the *Ae. koreicus* samples collected in the invaded areas. *Aedes koreicus* populations of Northern Italy and Slovenia were distributed mainly along the second axis and were strongly interconnected.

Similar results were obtained plotting the DAPC (Fig 2B). KO21 was separated more from the other invasive populations; these are distributed along Axis1, which explained most of the variance (68.0%). The cluster of populations from North-Eastern Italy (i.e., BL11, VI21, VI22), AT21, and SL21 could be also appreciated along Axis 2; in addition, DAPC demonstrated a relationship among populations from the North-Central area of Italy (CO21, SO21, SO22, TR22, FO21, FO22), except for BS21, whose individuals are partially shared among the two genetic clusters.

To allow a clearer visualization, PCA and DAPC were repeated only on the six populations considered to test overwintering (see Fig 2C and 2D). Despite the sharing of some individuals, that can be appreciated mostly in the PCA plot (Fig 2C), populations collected in the same localities in different years clustered together. This result was particularly evident among the individuals from Sondrio (SO21, SO22).

The Bayesian analysis of populations of *Ae. koreicus* was performed in STRUCTURE to evaluate genetic structuring and co-ancestry. A total of 11 populations collected in 2021 were analyzed since only samples collected in 2021 were considered for Vicenza, Sondrio, and Fonteno populations. STRUCTURE computed the probability of each specimen to be part of a specific cluster K (representative of the likely ancestral population); the total number of K that better describe the populations was deduced according to Evanno ΔK values (S6 Table). K = 2 resulted as the most representative co-ancestry scenario, followed by K = 3. Because of the genetic patterns observed in previous analyses, both co-ancestry representations were taken in consideration (K = 2 and K = 3).

K = 2 resulted from the genetic separation of individuals of KO21 from the specimens collected in Northern Italy and Slovenia (Fig 3 and S7 Table). Specifically, K2 was associated with KO21 (98.9%), while K1 described the invasive populations (probability of assignment to K1 >89% for all the invasive populations). This co-ancestry representation was already

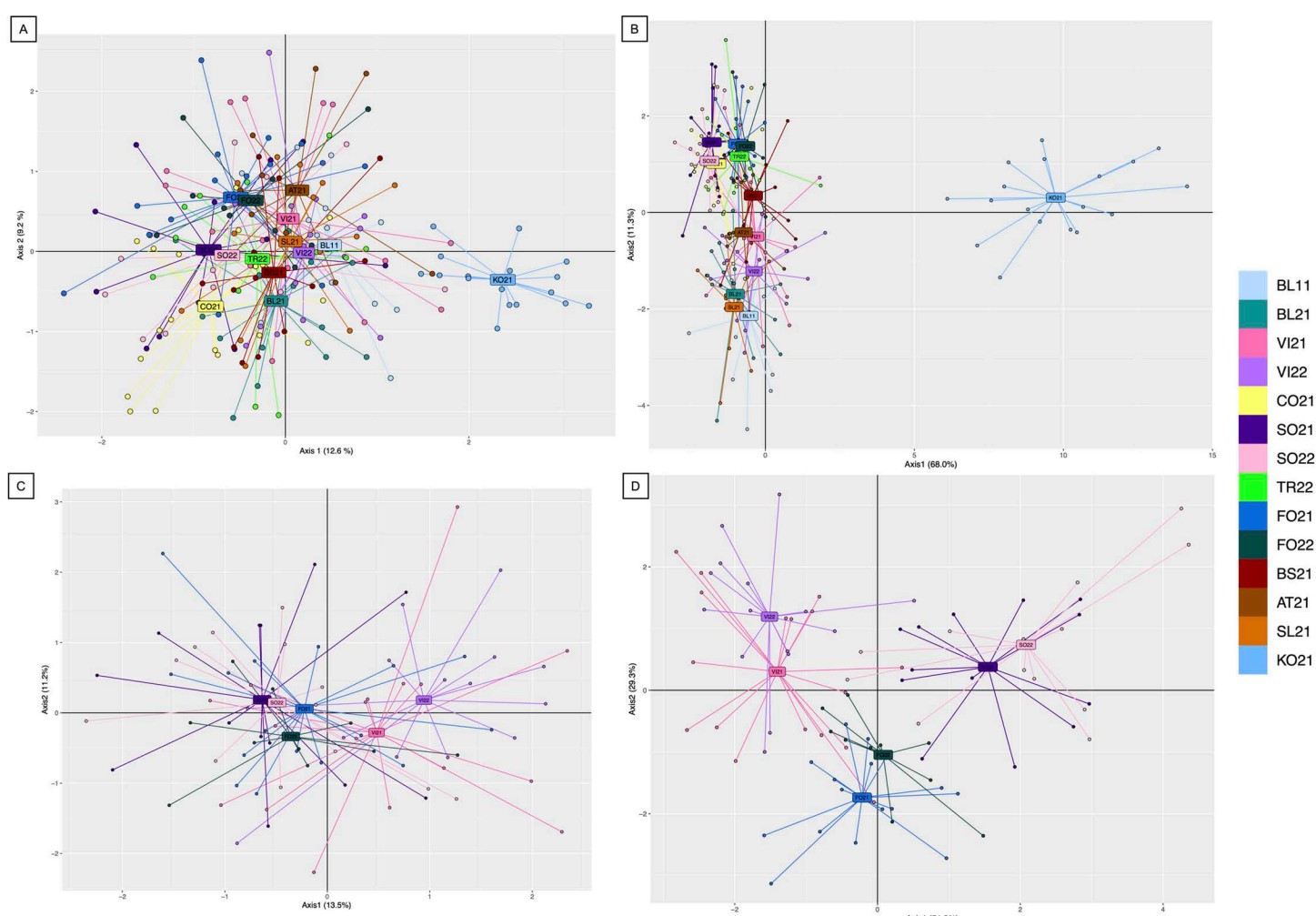

**Fig 2. PCA and DAPC plots of the populations of *Ae. koreicus*.** (A) PCA plot of all the 14 populations tested; the first two axes were represented here, explaining in the complex 21.8% of the genetic variation. (B) DAPC plot of all the 14 populations analysed; also in this case, the first two axes, which explained 79.3% of the variance associated to the genetic data. (C) PCA plot of the three couples of populations tested for overwintering, i.e., VI21-VI22, SO21-SO22, FO21-FO22; the first two axes were represented, explaining in the complex the 24.7% of the genetic variation. (B) DAPC plot of the three couples of populations tested for overwintering; the first two axes are represented, which explains the 80.6% of the variance associated with the genetic data. For all plots, the color legend is reported on the right.

highlighted by $F_{ST}P$ values and in the PCA and DAPC plots and confirms that KO21 does not represent the ancestral populations observed for *Ae. koreicus* collected in Italy and Slovenia.

The high ΔK observed for K = 2 is likely the consequence of elevated differentiation subsisting among the populations from the ROK and those from the invaded areas. Despite being useful for excluding KO21 as the original source of the specimens collected from disparate areas of Italy and Slovenia, the co-ancestry representation of K = 2 did not provide any clues about the structure of Italian and Slovenian populations. For this reason, an additional lineage was taken into consideration, K = 3. The separation of the population from ROK was evident (Fig 3), with KO21 only associated to K2 (97.8%) (S8 Table). Noteworthy, the introduction of an additional lineage allowed to better resolve the genetic structure of *Ae. koreicus* populations from Northern Italy and Slovenia. BL11, BL21, VI21, and SL21 populations were associated with K1, while CO21, SO21, TR22, and FO21 belonged to K3 (S8 Table). Among the first group of populations, the result of VI21 should be mentioned: individuals of this population,

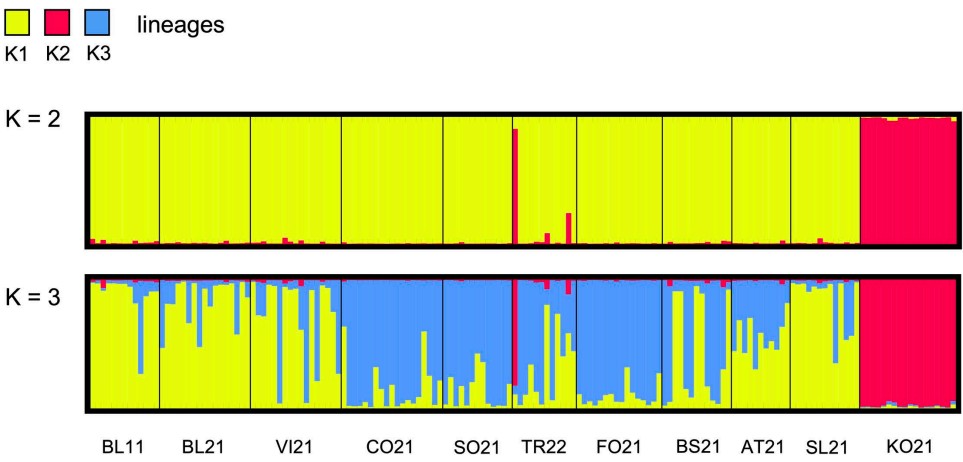

**Fig 3. Bayesian analyses of the genetic structure of the 11 populations of *Ae. koreicus*.** The co-ancestry distribution of the most probable scenarios was represented (K = 2, K = 3). The lineages legend is shown at the top, while the populations of *Ae. koreicus* included in the analysis are shown at the bottom.

even if mainly belonging to K1, were weakly distributed in K3 (K1: 69.7%; K3: 29.0%). Moreover, AT21 and BS21 were almost equally distributed among K1 and K3, with a slightly higher probability associated with K1 and K3, respectively (S8 Table).

## Demographic inferences on the invasive patterns of *Ae. koreicus* in Northern Italy and Slovenia

The preliminary demographic investigation of invasive patterns of the Korean bush mosquito in Northern Italy and Slovenia was tested taking into consideration ten disparate populations, i.e., BL11 together with other nine populations collected in the invaded area, namely BL21, VI21, CO21, SO21, TR22, FO21, BS21, AT21, SL21. The focus of the present analysis was the demography of *Ae. koreicus* in Northen Italy and Slovenia.

Based on previous genetics results, the most probable hypothesis was that the populations of *Ae. koreicus* dispersed in Italy and Slovenia were derived from the same original and, by now, still unidentified population (NA) which was first detected in Belluno district in 2011. Nine different scenarios, addressing this hypothesis, were tested (S1 Fig). Scenarios were different in terms of the derivation pathways of the Italian and Slovenian population which were rearranged on the basis of genetic relationships deduced from previous analyses and on the occurrence records.

Given the concordance of simulated dataset and logistic regression of posterior probabilities, scenario 4 was the most descriptive for the populations of *Ae. koreicus* (Fig 4). Particularly, the posterior probability associated to this scenario is equal to ~80% for all the test datasets, with a final value of 0.808 and a confidence interval of 0.689-0.927 (Fig 4A; see S9 Table for further details about posterior probabilities of all scenarios). Type I and type II errors associated to Scenario4 were 0.621 and 0.169, respectively.

## Discussion

The present study aims to investigate for the first time the dispersal patterns and genetic structure of populations of *Ae. koreicus* in Northern Italy, a region where this alien mosquito species has established in the last decade and where is predicted to further expand [5,6,11,27]. Understanding these patterns is crucial for effective future control interventions necessary

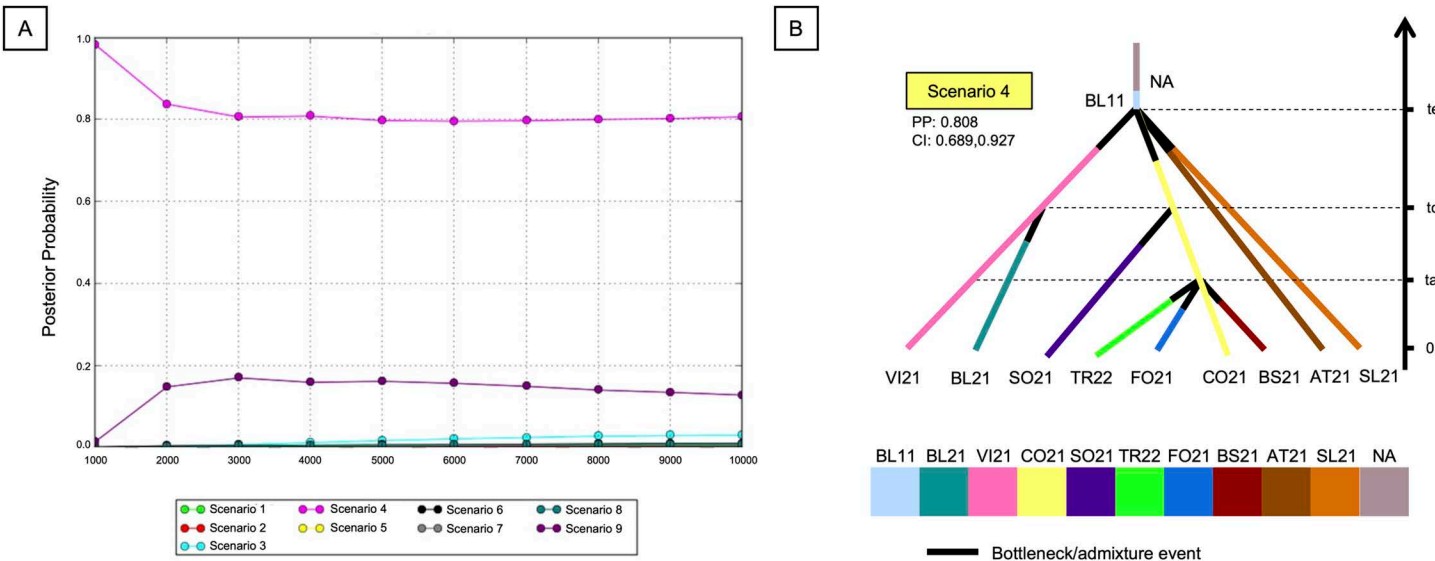

**Fig 4. Demographic expansion of *Ae. koreicus* in Northern Italy and Slovenia.** (A) Logistic regression of the posterior probabilities of all the 9 scenarios designed to test the demography of the populations of *Ae. koreicus;* the posterior probabilities were computed by DIYABC, and Scenario 4 (in dark pink) resulted as the most probable in all simulated datasets. (B) Graphical representation of Scenario 4, i.e., the most probable scenario for the demographic expansion of *Ae. koreicus* in Northern Italy and Slovenia.

to mitigate its spread and potential health risk. To achieve this, specific genetic microsatellite markers were used to analyze 14 disparate populations of *Ae. koreicus*, with samples from Italy compared to specimens from Slovenia, and one population from the Republic of Korea (ROK). Since the primary focus was the invasive situation of Northern Italy, populations representative of eight localities spanning the pre-alpine region collected in 2021-2022 were considered in the analyses. Specimens collected in Belluno in 2011 were also included, as they represent the first population of *Ae. koreicus* detected in Italy [4]. This was critical for understanding whether its expansion or multiple introductions led to the current distribution.

Principal results of our analyses demonstrated *Ae. koreicus* has fully adapt to Northern Italy, with populations exhibiting genetic homogeneity across the region and likely originating by the expansion of the 2011 Belluno population. Indeed, the population from Belluno collected in 2011 displayed low genetic differentiation with most of the other Italian populations. Bayesian analyses supported this hypothesis, showing that individuals introduced in Belluno spread into Slovenia and westward across Northern Italy, founding new invasive populations.

In addition, the presence of overwintering populations of *Ae. koreicus* in Italy was confirmed in several districts (i.e., Vicenza, Sondrio, and Fonteno), where increasing inbreeding levels, clustering across consecutive years, and low genetic differentiation suggested the presence of stable populations. This is in concordance with previous observational data and is also supported by the finding of overwintering individuals in Hungary [58–60]. These results suggest that the biology of this mosquito species allows individuals to easily adapt to new environments, ensuring a successful invasion.

Despite the elevated genetic similarities, evidence of patterns of regional clustering in the invaded area were found. Populations in the Lombardy region (i.e., Como, Sondrio, Trescore Balneario, and Fonteno) formed a distinct cluster from the populations collected in the Veneto region (both populations from Belluno, and Vicenza), which grouped with Slovenia. Interestingly, the population from Brescia (Lombardy region), first recorded in 2021 [5], was not

at HWE equilibrium (excess of heterozygous individuals) and showed evidence of admixture between the Lombardy and the Veneto regions (STRUCTURE and DAPC analyses). This evidence suggests that Brescia may be the point of convergence for migrating individuals from different regions. The fusion of local distinct populations was already described in endemic species of mosquitoes. For example, a similar phenomenon was detected for the sibling species *Aedes japonicus* during the colonization of Germany and USA [61–63]. In addition, local admixture was already suggested for Italian *Ae. koreicus* populations by analyzing the mitochondrial haplotype distribution of specimens from Northeastern Italy [24].

Populations such as Trescore Balneario (in the Bergamo district) and Asti exhibited unique patterns. Trescore Balneario had high allelic richness and private alleles, indicating possible multiple introductions, while a small proportion of individuals showed genetic similarity to the ROK population, hinting at secondary introductions. Asti, located in the Piedmont region and farthest from Belluno, grouped with the Veneto cluster. This genetic similarity might be easily explained by considering the occurrence of a passive man-mediated dispersal of invasive species, which is a common phenomenon during mosquito invasion processes [23]. For instance, this dispersion pathway was described for *Ae. japonicus* both in Europe and in the USA [23,64–66].

The strong genetic differentiation among the ROK population and all invasive populations indicates that the Italian and Slovenian populations likely did not originate from the Korean population included in this study. Further samplings in distinct endemic regions, e.g., Russia and China, or from different areas of the ROK, are needed to pinpoint the original source of the invasion of Northern Italy.

These genetic data reflect anyhow the division of *Ae. koreicus* into two distinct morphotypes [9–11], the one detected in the ROK mainland, visible in the KO21 population, and the other one described in Jeju Island, characterizing all the individuals collected in Italy and Slovenia.

The cumulative results and the single introduction and expansion model proposed are robust, but some methodological limitations must be accounted. Microsatellites are less powerful than Single Nucleotide Polymorphisms (SNPs) in identifying fine-scale patterns of recent divergence and local adaptation [67–69]. Future studies incorporating SNP markers will provide higher-resolution data, enabling a deeper understanding of the invasion history and evolutionary dynamics of *Ae. koreicus*.

## Conclusion

In conclusion, the present study provided novel data and results with a possible reconstruction of the pattern of dispersal of *Ae. koreicus* in Northern Italy, supporting the hypothesis of a spread of an original population firstly introduced near Belluno that expanded from the east to the west of the Alpine region. Anyway, signs of multiple introduction events have been described at least in one locality, Trescore Balneario. Further studies will properly identify the source of the introduction of *Ae. koreicus* in Italy. Monitoring activity should remain constant, and possibly more comprehensive and integrated actions across regional boundaries should be implemented to avoid further expansion of this invasive mosquito species.

## Supporting information

**S1 File. Genotypes of *Ae. koreicus* mosquitoes.** For each of the 10 *loci* included in the population genetics analyses, genotypes of single mosquitoes are reported rounded to the integer. The length of the repetitive motif characterizing each *locus* is reported too.
(XLSX)

**S1 Fig. Possible demographic scenarios of the expansion of Ae. koreicus in Northern Italy and Slovenia.** Representation of the scenarios designed and tested in DIYABC are reported. The scenario resulting as most representative (Scenario 4) on the basis of Bayesian computation is highlighted in yellow; for the latter, the posterior probability (PP) and confidence interval (CI) are reported.
(PDF)

**S2 Fig. Linear correlation between genetic distance (FSTP) and geographic distance (GGD).** This is the result of 1000 permutations, performed to assess the possible isolation by distance in *Ae. koreicus* populations in Italy and Slovenia.
(PDF)

**S1 Table. Details of the 414 individuals of Ae. koreicus analysed in the present work.** Specifically, for each specimen are reported: population they belong to (Pop), individual identification code (ID), sex, date and stage of collection, the collection site and its coordinates (both as decimal and UTM).
(XLSX)

**S2 Table. List of microsatellite markers and primer pairs used in the present study.** For each locus are specified: the repeat motifs; forward (F) and reverse (R) primer sequences, and fluorophores used as markers. The organization of the multiplex assay is also reported.
(XLSX)

**S3 Table. Relatedness analysis of Ae. koreicus individuals collected in the same breeding site.** Both the results of SPAGeDi and ML-relate analysis are presented. The final degree of relation, i.e., full sibling (FS), half sibling (HS), and unrelated (U) has been established according to both outputs and is reported in the column "Relationship".
(XLSX)

**S4 Table. Basic statistics of intra-population variability computed for each *locus* in all 14 populations of Ae. koreicus.** Particularly, are reported: number of individuals (N), number of alleles (Na), number of private alleles (Np) and their frequencies in the populations, observed (Ho) and expected (He) heterozygosities, Inbreeding coefficient ($F_{IS}$), and Polymorphic Information Content (PIC).
(XLSX)

**S5 Table. 95% confidence intervals related to FSTP values.** These intervales were derived from the computation of 10000 bootstrap.
(XLSX)

**S6 Table. Table with the likelihood values associated to each cluster (K).** The latter was produced as output of STRUCTURE HARVESTER, where are reported for each of the K tested: the number of repetitions (Reps), the mean value of the likelihood distribution (Mean LnP(K)) and the relative Standard Error, the mean of first rate of change of the likelihood K distribution (Ln'(K)), the mean absolute value of the 2nd order rate of change of the likelihood K distribution (|Ln"(K)|), the probability of K distribution obtained as: mean(|L"(K)|)/sd(L(K)) (ΔK). The two most probable co-ancestry distribution (K = 2, K = 3) are highlighted in bold.
(PDF)

**S7 Table. Summary of the K probability for K = 2 (obtained with CLUMPP).** The highest value is highlighted for each population of *Ae. koreicus* here analysed.
(PDF)

**S8 Table. Summary of the K probability for K = 3 (obtained with CLUMPP).** The highest value is highlighted for each population of *Ae. koreicus* here analysed.
(PDF)

**S9 Table. Posterior probabilities and 95% confidence intervals computed by ABC for all the 9 tested scenarios.** Highest values of posterior probability are highlighted, and resulted all associated to Scenario 4 in all the tested datasets (n).
(PDF)

## Acknowledgements

The authors would like to thank the citizens for their support and availability during field collections.

The opinions or assertions contained herein are the private views of the authors and are not to be construed as official or reflecting the true views of the U.S. Department of the Army, U.S. Department of Defense, or the U.S. Government. This work was prepared in accordance with Title 17, U.S.C., §101 that defines a U.S. Government work as work prepared by a military service member of the U.S. Government (TA Klein) as part of that person's official duties.

## Author contributions

**Conceptualization:** Laura Soresinetti, Sara Epis, Paolo Gabrieli.

**Data curation:** Laura Soresinetti, Giovanni Naro, Irene Arnoldi.

**Funding acquisition:** Claudio Bandi, Sara Epis, Paolo Gabrieli.

**Investigation:** Laura Soresinetti, Giovanni Naro.

**Resources:** Laura Soresinetti, Irene Arnoldi, Andrea Mosca, Katja Adam, Heung Chul Kim, Terry A. Klein, Francesco Gradoni, Fabrizio Montarsi, Sara Epis, Paolo Gabrieli.

**Supervision:** Claudio Bandi, Sara Epis, Paolo Gabrieli.

**Writing – original draft:** Laura Soresinetti, Giovanni Naro.

**Writing – review & editing:** Claudio Bandi, Sara Epis, Paolo Gabrieli.

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
