## [Decision Letter · Decision Letter 0]

2 Dec 2024

PNTD-D-24-01498The genetic trail of the invasive mosquito species Aedes koreicus from the east to the west of Northern ItalyPLOS Neglected Tropical DiseasesDear Dr. Gabrieli, Thank you for submitting your manuscript to PLOS Neglected Tropical Diseases. After careful consideration, we feel that it has merit but does not fully meet PLOS Neglected Tropical Diseases's publication criteria as it currently stands. Therefore, we invite you to submit a revised version of the manuscript that addresses the points raised during the review process. Please submit your revised manuscript within 30 days Jan 30 2025 11:59PM. If you will need more time than this to complete your revisions, please reply to this message or contact the journal office at plosntds@plos.org. Please include the following items when submitting your revised manuscript: * A rebuttal letter that responds to each point raised by the editor and reviewer(s). You should upload this letter as a separate file labeled 'Response to Reviewers '. This file does not need to include responses to any formatting updates and technical items listed in the 'Journal Requirements' section below. * A marked-up copy of your manuscript that highlights changes made to the original version. You should upload this as a separate file labeled 'Revised Manuscript with Track Changes '. * An unmarked version of your revised paper without tracked changes. You should upload this as a separate file labeled 'Manuscript '. If you would like to make changes to your financial disclosure, competing interests statement, or data availability statement, please make these updates within the submission form at the time of resubmission. Guidelines for resubmitting your figure files are available below the reviewer comments at the end of this letter. We look forward to receiving your revised manuscript. Kind regards,Clarence Mang'era, PhDGuest EditorPLOS Neglected Tropical Diseases Paul MirejiSection EditorPLOS Neglected Tropical Diseases

Shaden Kamhawi

co-Editor-in-Chief

Paul Brindley

co-Editor-in-Chief

**Additional Editor Comments :** Carefully determine which data to present in the main manuscript for exhuastive discussion while placing the remaining data in the supplementary section for clarity and conciseness. **Journal Requirements:**

At this stage, the following Authors/Authors require contributions: Laura Soresinetti, Giovanni Naro, Irene Arnoldi, Andrea Mosca, Katja Adam, Heung Chul Kim, Terry A. Klein, Francesco Gradoni, Fabrizio Montarsi, Claudio Bandi, Sara Epis, and Paolo Gabrieli. Please ensure that the full contributions of each author are acknowledged in the "Add/Edit/Remove Authors" section of our submission form.

Potential Copyright Issues:

i) Figure 1. Please (a) provide a direct link to the base layer of the map (i.e., the country or region border shape) and ensure this is also included in the figure legend; and (b) provide a link to the terms of use / license information for the base layer image or shapefile. We cannot publish proprietary or copyrighted maps (e.g. Google Maps, Mapquest) and the terms of use for your map base layer must be compatible with our CC BY 4.0 license.

5) In the online submission form, you stated that "raw genotypes of individual mosquitoes included in the analyses of the present study. that can be accessed on Zenodo at the following link: doi.org/10.5281/zenodo.12800346." We noticed that the data will be made publicly available on July 31,2025. We strongly recommend all authors deposit their data before acceptance, as the process can be lengthy and hold up publication timelines. Please note that, though access restrictions are acceptable now, your entire minimal dataset will need to be made freely accessible if your manuscript is accepted for publication. This policy applies to all data except where public deposition would breach compliance with the protocol approved by your research ethics board. If you are unable to adhere to our open data policy, please kindly revise your statement to explain your reasoning and we will seek the editor's input on an exemption.

6) Thank you for stating that 'All data and results of the analyses are available within the manuscript or in the Supporting Information files.' Please confirm at this time whether or not your submission contains all raw data required to replicate the results of your study. Authors must share the “minimal data set” for their submission. PLOS defines the minimal data set to consist of the data required to replicate all study findings reported in the article, as well as related metadata and methods (https://journals.plos.org/plosone/s/data-availability#loc-minimal-data-set-definition). For example, authors should submit the following data: - The values behind the means, standard deviations and other measures reported; - The values used to build graphs; - The points extracted from images for analysis. Authors do not need to submit their entire data set if only a portion of the data was used in the reported study. If your submission does not contain these data, please either upload them as Supporting Information files or deposit them to a stable, public repository and provide us with the relevant URLs, DOIs, or accession numbers. For a list of recommended repositories, please see https://journals.plos.org/plosone/s/recommended-repositories. If there are ethical or legal restrictions on sharing a de-identified data set, please explain them in detail (e.g., data contain potentially sensitive information, data are owned by a third-party organization, etc.) and who has imposed them (e.g., an ethics committee). Please also provide contact information for a data access committee, ethics committee, or other institutional body to which data requests may be sent. If data are owned by a third party, please indicate how others may request data access.

**Comment to the Authors:**

Please note that one of the reviews was submitted as an attachment file.

**Reviewers' comments:**Reviewer's Responses to Questions

**Key Review Criteria Required for Acceptance?**

**Methods**

-Are the objectives of the study clearly articulated with a clear testable hypothesis stated?

-Is the study design appropriate to address the stated objectives?

-Is the population clearly described and appropriate for the hypothesis being tested?

-Is the sample size sufficient to ensure adequate power to address the hypothesis being tested?

-Were correct statistical analysis used to support conclusions?

-Are there concerns about ethical or regulatory requirements being met?

Reviewer #1: A lot of data analysis were done but mostly not necessary to answer the research objectives. I suggest to add additional samples from endemic countries such as China, Russia, other sites in Korea. The information about the number of samples per population should also be mentioned in the methodology. Sample size is sufficient if the objectives is to know the population genetic structure of the Ae. kroeius. However, it was mentioned that the objectives "we aimed at elucidating the processes driving its invasion dynamics in Italy, and potentially in the rest of Europe.” So in this regard, i suggest to add samples from endemic countries and other neighboring European countries. I suggest to perform UPGMA dendrogram to determine genetic groups and also analyze the migration rate between the sites using Geneclass. Testing the isolation by distance using mantel test is also suggested. There are many unnecessary data analysis performed but it doesn't answered well your research objectives. I also suggest to determine if the genetic distance is affected by geographic distance so a spatial autocorrelation test is also suggested.

Reviewer #2: yes to all above mentioned questions

Reviewer #3: (No Response)

**Results**

-Does the analysis presented match the analysis plan?

-Are the results clearly and completely presented?

-Are the figures (Tables, Images) of sufficient quality for clarity?

Reviewer #1: The results are not completely presented and most results are not necessarily main and i suggest to put it as supplementary file. please see further comments in the attached file

Reviewer #2: yes to all above mentioned questions

Reviewer #3: (No Response)

**Conclusions**

-Are the conclusions supported by the data presented?

-Are the limitations of analysis clearly described?

-Do the authors discuss how these data can be helpful to advance our understanding of the topic under study?

-Is public health relevance addressed?

Reviewer #1: The limitations of the study was mentioned however this limitation is necessary to answer the research objective so i suggest to add aditional samples or revise your research objectives.

Reviewer #2: Discussion: it would be worth including a paragraph about the methodological limitations as after all, the amount of SSR markers investigated in this study is quite limited.

Reviewer #3: (No Response)

**Editorial and Data Presentation Modifications?**

Reviewer #1: Minor Revision

Reviewer #2: Below, some minor comments to the paper:

- Line 141: it is clear from your table but I would mention 10 collection sites in this sentence instead of 14, and specify that for 4 collection sites, you include a temporal component to the investigation (which I highly appreciate).

- Line 157: could you specify if you visited the collection sites multiple times over year- and if so at which intervals

- Line 221: I do not understand this sentence: could you please rephrase

- Line 353: add a reference to the sentence

- Line 412 and 415: once indicated 10000, once 1000; a typo?

- Line 421-423: could you clarify this statement

- Line 430: did you also check differences in rare alleles between populations collected at the same sites over multiple years?

- Line 530: referring to your first table, samples from tr22 were collected in 2022 and not 2021

Reviewer #3: (No Response)

**Summary and General Comments**

Reviewer #1: Review

L54-55 Do you mean 14 populations in total, 13 from Italy and Slovenia and one from Korea? Please confirm and revise if necessary.

Table 1. please put the meaning of abbreviation, N under the table.

For table 4, I suggest to put it as supplementary file and make it more organized, it is too messy to look at.

In the methodology, I suggest to include a supplementary table of the list of the microsatellites and their reference journal.

L562-567 are redundant and was mentioned already previously.

L581-583 I agree that including a population from the endemic region is important however samples were only collected from one site and I suggest to include representative samples from other sites and or other endemic countries so you can compare the genetic structure and relatedness of the samples from Italy.

L587-588 This is a strong claim that the samples are not related from ROK since you only sampled few individuals and from a single site which is can also be a factor.

I suggest to deposit your dataset in publicly available database for example vectorbase.org

The authors have done extensive data analysis however it seems that some data exploited are not necessarily needed to answer their objective to “population

genetics of Ae. koreicus, with a particular focus on the populations dispersed

throughout the pre-alpine area of Northern Italy. Through comprehensive population

genetic analyses, we aimed at elucidating the processes driving its invasion

dynamics in Italy, and potentially in the rest of Europe.”

If these is the main objective it is necessary to sample individual mosquitoes from native countries of Ae. Koreicus such as in China, Russia and from neighboring European countries. The results does not wholly answered the objectives however it showed preliminary data supporting the genetic structure of the mosquitoes in Italy, I suggest to revise the objectives and focus mainly on the population genetics of Ae. Koreicus in Italy and remove the samples from Korea in all the data analysis.

Additional data analysis are needed to answer the research questions. The discussion part should be improved more by discussing more about how their study is novel and how it answer research gaps. The novelty and importance of their study was not highlighted.

Reviewer #2: The present research employs recently described SSR markers to contribute unraveling the introduction pathways and invasive potential, as well as colonization pattern of Aedes koreicus in northern Italy and parts of Slovenia. I must admit that I have not much to comment as the paper is already well-developed, complete (including very clear figures), fluently written, with a clear and logical structuring. The methodology is thoroughly explained, and the paper contributes to the field of invasive species colonization patterns. I therefore recommend the paper for publication in PLOS Neglected Tropical Diseases, also since it fits the journal scope. Congratulations on this study.

Reviewer #3: (No Response)

PLOS authors have the option to publish the peer review history of their article (what does this mean? ). If published, this will include your full peer review and any attached files.

**Do you want your identity to be public for this peer review?** For information about this choice, including consent withdrawal, please see our Privacy Policy .

Reviewer #1: **Yes: ** Maria Angenica F. Regilme

Reviewer #2: No

Reviewer #3: No

**Figure resubmission:**While revising your submission, please upload your figure files to the Preflight Analysis and Conversion Engine (PACE) digital diagnostic tool, https://pacev2.apexcovantage.com/. PACE helps ensure that figures meet PLOS requirements. To use PACE, you must first register as a user. Registration is free. Then, login and navigate to the UPLOAD tab, where you will find detailed instructions on how to use the tool. If you encounter any issues or have any questions when using PACE, please email PLOS at figures@plos.org. Please note that Supporting Information files do not need this step. If there are other versions of figure files still present in your submission file inventory at resubmission, please replace them with the PACE-processed versions. **Reproducibility:**To enhance the reproducibility of your results, we recommend that authors of applicable studies deposit laboratory protocols in protocols.io, where a protocol can be assigned its own identifier (DOI) such that it can be cited independently in the future. Additionally, PLOS ONE offers an option to publish peer-reviewed clinical study protocols. Read more information on sharing protocols at https://plos.org/protocols?utm_medium=editorial-email&utm_source=authorletters&utm_campaign=protocols

---

## [Decision Letter · Decision Letter 1]

25 Feb 2025

Dear Professor Gabrieli,

We are pleased to inform you that your manuscript 'The genetic trail of the invasive mosquito species Aedes koreicus from the east to the west of Northern Italy' has been provisionally accepted for publication in PLOS Neglected Tropical Diseases.

Best regards,

Clarence Mang'era, PhD

Guest Editor

Paul Mireji

Section Editor

Shaden Kamhawi

co-Editor-in-Chief

Paul Brindley

co-Editor-in-Chief

Please address the outstanding reviewer concerns and emphasize the significance of this study in vector control, particularly by discussing the potential public health implications if this species becomes a probable disease vector in the future. Additionally, clarify any potential contradictions between the retained datasets from the Republic of Korea and Slovenia to ensure consistency.

Reviewer's Responses to Questions

**Key Review Criteria Required for Acceptance?**

**Methods**

-Are the objectives of the study clearly articulated with a clear testable hypothesis stated?

-Is the study design appropriate to address the stated objectives?

-Is the population clearly described and appropriate for the hypothesis being tested?

-Is the sample size sufficient to ensure adequate power to address the hypothesis being tested?

-Were correct statistical analysis used to support conclusions?

-Are there concerns about ethical or regulatory requirements being met?

Reviewer #1: The objectives of the study as stated as

"In the present study, microsatellite markers were used to perform population genetics of Ae. koreicus, with a particular focus on the populations dispersed throughout the pre-alpine area of Northern Italy. Through comprehensive population genetic analyses, we aimed at elucidating the genetic relationship and the invasion dynamics in Italy." were clearly stated but the authors seem to lack a clear testable hypothesis statement. Though the samples analyzed are enough with additional samples from Slovenia, it is unclear why samples other than Italy were added, since the objectives stated that they will focus on populations on pre-alpine area of Northern Italy. It would be suggested to revise the objectives and hypothesis to fit more perfectly the study design. Since you wanted to know the genetic structure of the mosquitoes and you have samples from Italy and Slovenia, as suggested previously migration rate test will be helpful to determine the migtration patterns, however the authors did not performed this and did not provide a valuable explanation with supporting information as rebuttal to the suggestion. The authors argued about PCA but it does not directly estimate migration rates or detect recent migrants which can be detected by analysis of migration using gene class. A more scientific justification on why the authors did not choosed to performed these suggested analysis will be very much appreaciated.

Reviewer #2: yes

**Results**

-Does the analysis presented match the analysis plan?

-Are the results clearly and completely presented?

-Are the figures (Tables, Images) of sufficient quality for clarity?

Reviewer #1: The figures looks blurred it might be from the compression of the files into pdf, so please double check.

Reviewer #2: yes

**Conclusions**

-Are the conclusions supported by the data presented?

-Are the limitations of analysis clearly described?

-Do the authors discuss how these data can be helpful to advance our understanding of the topic under study?

-Is public health relevance addressed?

Reviewer #1: In l567-577, though the authors mentioned about the limitations of microsatellites and the suggestion of using SNPs, i suggest to also add justification on the use of microsatellites not just saying the results were robust, for example previous studies that used microsatellites or the principle of microsatellites. In the conclusion, it would be better if the authors propose a more specific plan on how their results can help in vector-control for example if this species will be a probable disease vector in the future. Authors can mentioned the public health importance of their study.

Reviewer #2: yes

**Editorial and Data Presentation Modifications?**

Reviewer #1: (No Response)

Reviewer #2: The authors have adequately addressed my comments and have thoroughly reviewed the discussion to comply with Reviewer 1's comments. Therefore, I recommend accepting the manuscript for publication in PLOS Neglected Tropical Diseases.

**Summary and General Comments**

Reviewer #1: The authors have done an extensive data analysis in trying to answer their research objectives. I suggest to highlight the public health significance of the research in the community.

Reviewer #2: (No Response)

PLOS authors have the option to publish the peer review history of their article (what does this mean? ). If published, this will include your full peer review and any attached files.

**Do you want your identity to be public for this peer review?** For information about this choice, including consent withdrawal, please see our Privacy Policy .

Reviewer #1: No

Reviewer #2: No

---

## [Editor Report · Acceptance letter]

Dear Professor Gabrieli,

We are delighted to inform you that your manuscript, "The genetic trail of the invasive mosquito species Aedes koreicus from the east to the west of Northern Italy," has been formally accepted for publication in PLOS Neglected Tropical Diseases.

Best regards,

Shaden Kamhawi

co-Editor-in-Chief

Paul Brindley

co-Editor-in-Chief
